# QCue: Queries and Cues for Computer-Facilitated Mind-Mapping

Ting-Ju Chen*
J. Mike Walker '66 Department
of Mechanical Engineering
Texas A&M University

Sai Ganesh Subramanian†
J. Mike Walker '66 Department
of Mechanical Engineering
Texas A&M University

Vinayak R. Krishnamurthy‡
J. Mike Walker '66 Department
of Mechanical Engineering
Texas A&M University

## ABSTRACT

We introduce a novel workflow, *QCue*, for providing textual stimulation during mind-mapping. Mind-mapping is a powerful tool whose intent is to allow one to externalize ideas and their relationships surrounding a central problem. The key challenge in mind-mapping is the difficulty in balancing the exploration of different aspects of the problem (breadth) with a detailed exploration of each of those aspects (depth). Our idea behind *QCue* is based on two mechanisms: (1) computer-generated automatic cues to stimulate the user to explore the breadth of topics based on the temporal and topological evolution of a mind-map and (2) user-elicited queries for helping the user explore the depth for a given topic. We present a two-phase study wherein the first phase provided insights that led to the development of our work-flow for stimulating the user through cues and queries. In the second phase, we present a between-subjects evaluation comparing *QCue* with a digital mind-mapping work-flow without computer intervention. Finally, we present an expert rater evaluation of the mind-maps created by users in conjunction with user feedback.

**Index Terms:** Human-centered computing—Visualization—Visualization techniques—Treemaps; Human-centered computing—Visualization—Visualization design and evaluation methods

## 1 INTRODUCTION

Mind-maps are widely used for quick visual externalization of one's mental model around a central idea or problem. The underlying principle behind mind-mapping is to provide a means for associative thinking so as to foster the development of concepts that both explore different aspects around a given problem (breadth) and explore each of those aspects in a detail-oriented manner (depth) [49]. The ideas in a mind-map spread out in a hierarchical/tree-like manner [35], which allows for the integration of diverse knowledge elements into a coherent pattern [8] to enable critical thinking and learning through making synaptic connections and divergent exploration [41, 56,77,78]. As a result, mind-maps are uniquely suitable for *problem understanding/exploration* prior to design conceptualization [8].

Problem exploration is critical in helping designers develop new perspectives and driving the search for solutions within the iterative process of identifying features/needs and re-framing the scope [53]. Generally, it requires a combination of two distinct and often conflicted modes of thinking: (1) logical, analytical, and detail-oriented, and (2) lateral, systems-level, breadth-oriented [40]. Most current efforts in computer-facilitated exploratory tasks focus exclusively on one of these cognitive mechanisms. As a result, there is currently a limited understanding of how this breadth-depth conflict can be addressed. Maintaining the balance between the breadth and depth of exploration can be challenging, especially for first-time users. For atypical and open-ended problem statements (that are

commonplace in design problems), this issue is further pronounced ultimately leading to creative inhibition and lack of engagement.

Effective and quick thinking is closely tied to the individual's imagination and ability to create associations between various information chunks [44]. Incidentally, this is also a skill that takes time to develop and manifest in novices. We draw from existing works [17, 22, 27, 29, 38, 60, 74, 79] that emphasize on stimulating reflection during exploration tasks. Quayle et al. [60] and Wetzstein et al. [74] indicate that the act of responding to questions can create several avenues for designers to reflect on their their assumptions and expand their field of view about a given idea. Adler et al. [3] found that asking questions in sketching activity keeps the participants engaged and reflecting on ambiguities. In fact, asking one question in turn raises a variety of other questions, thereby bringing out more ideas from the user's mind [17]. Goldschmidt [29] further demonstrated that exposing designers to text can lead to higher originality during idea generation.

Our approach is informed by the notion of *reflection-in-design* [60, 74], that takes an almost Socratic approach to reason about the design problem space through question-based verbalization. The premise is that cognitive processes underlying mind-mapping can be enriched to enable an iterative cycle between exploration, inquiry, and reflection. We apply this reasoning in a digital setting where the user has access to vast knowledge databases. Our key idea is to explore two different ways in which such textual stimuli can be provided. The first is through a simple mechanism for query expansion (i.e. asking for suggestions) and followed by means for responding to computer-generated stimuli (i.e. answering questions). Based on this, we present a workflow for mind-mapping wherein the user, while adding and connecting concepts (*exploration*), can also query a semantic database to explore related concepts (*inquiry*) and build upon those concepts by answering questions posed by the mind-mapping tool itself (*reflection*). Our approach is powered by ConceptNet [66], a semantic network that contains a graph-based representation with nodes representing real-word concepts as natural language phrases (e.g. bring to a potluck, provide comfort, etc.), and edges representing semantic relationships. Using related entries to a given concept and also the types of relationships, our work investigates methods for textual stimulation for mind-mapping.

### 1.1 Contributions

We make three contributions. First, we present a novel workflow — *QCue* — that uses the relational ontology offered by ConceptNet [66] to create mechanisms for cognitive stimulus through automated questioning with idea expansion and proactive user query. Second, we present an adaptive algorithm to facilitate the breadth-depth balance in a mind-mapping task. This algorithm analyzes the temporal and topological evolution of a given mind-map and generates questions (cues) for the user to respond to. Finally, to showcase the reflection-in-design approach, we conduct a between-subjects user study and present a comparative evaluation of *QCue* with a digital mind-mapping workflow without our algorithm (henceforth in the paper, we will refer to this as traditional mind-mapping or TMM). The inter-rater analysis of user-generated mind-maps and the user feedback demonstrates the efficacy of our approach and also reveals new directions for future digital mind-mapping tools.

---
*e-mail: carol0712@tamu.edu

†e-mail: sai3097ganesh@tamu.edu

‡e-mail: vinayak@tamu.edu

## 2 Related Works

### 2.1 Problem Exploration in Design

Problem exploration is the process that leads to discovery of opportunities and insights that drives the innovation of products, services and systems [18]. Silver et al. [31] underscore the importance of problem-based learning for students to identify what they need to learn in order to solve a problem. Most current methods in early design are generally focused on increasing the probability of coming up with creative solutions by promoting divergent thinking. For instance, brainstorming specifically focuses on the quantity of ideas without judgment [6, 48, 57]. There are many other popular techniques such as SCAMPER [51], C-Sketch [64], and morphological matrix [80], that support the formation of new concepts through modification and re-interpretation of rough initial ideas. This however, also leads to design fixation toward a specific and narrow set of concepts thereby curtailing the exploration process. In contrast, mind-mapping is a flexible technique that can help investigate problem from multiple points of view. In this paper, we use mind-mapping as means for problem exploration, which has been proven to be useful for reflection, communication, and synthesis during idea generation [33, 50]. The structure of mind-maps thus facilitates a wide-range of activities ranging from note-taking to information integration [20] by highlighting the relationships between various concepts and the organization of topic-oriented flow of thoughts [55, 61].

### 2.2 Computer-Based Cognitive Support

There have been significant efforts to engage and facilitate ones' critical thinking and learning by using digital workflows through pictorial stimuli [30, 32, 71], heuristic-based feedback generation [72], text-mining [46, 65, 67] and speech-based interfaces [19]. Some works [13, 26] have also used gamification as a means to engage the user in the idea generation process. Specifically in engineering design and systems engineering, there are a number of computer systems that support user's creativity during design conceptualization [4, 58, 62, 70]. These are, however, targeted toward highly technical and domain-specific contexts.

While there are works [2, 24, 43, 73] that have explored the possibility of automatic generation of mind-maps from speech and texts, little is known in terms of how additional computer support will affect the process of creating mind-maps. Works that consider computer support in mind-mapping [7, 25] have evaluated numerous existing mind-mapping software applications and found that pen-and-paper and digital mind-mapping proves to have different levels of speed and efficiency analyzing various factors like user's intent, ethnography, nature of collaboration. Of particular relevance are works by Kerne's group on curation [34, 36, 47] and web-semantics [59, 76] for information based-ideation. While these works are not particularly aimed at mind-mapping as a mode of exploration, they share our premise of using information to support free-form visual exploration of ideas.

Recent work by Chen et al. [12] studies collaboration in mind-mapping and offer some insight regarding how mind-maps evolve during collaboration. They further proposed a computer as a partner approach [13], where they demonstrate human-AI partnership by posing mind-mapping as a two-player game where the human and the AI (intelligent agent) take turns to add ideas to a mind-map. While an exciting prospect, we note that there is currently little information regarding how intelligent systems could be used for augmenting the user's cognitive capabilities for free-form mind-mapping without constraining the process. Recent work by Koch et al. [38] proposed *cooperative contextual bandits* (CCB) that provides cognitive support in forms of suggestions (visual materials) and explanations (questions to justify the categories of designers' selections from search engine) to users during mood board design tasks. While CCB treats questions as means to justify designers'

focus and adapt the system accordingly, we emphasize the associative thinking capability brought by questions formed with semantic relations.

### 2.3 Digital Mind-Mapping

Several digital tools [75] have been proposed to facilitate mind-mapping activity. However, to our knowledge, these tools contribute little in computer-supported cognitive assistance and idea generation during such thinking and learning process. They focus on making the operations of constructing maps easier by providing features to users such as quickly expand the conceptual domain through web-search, link concepts to on-line resources via URLs (uniform resource locators) and interactive map construction. Even though those tools have demonstrated advantages over traditional mind-mapping tasks [25], mind-map creators can still find it challenging due to several following reasons: inability to recall concepts related to a given problem, inherent ambiguity in the central problem, and difficulty in building relationships between different concepts [5, 68]. These difficulties often result in an unbalanced idea exploration resulting in either too broad or too detail-oriented mind-maps. In this work, we aim to investigate computational mechanisms to address this issue.

## 3 Phase I: Preliminary Study

Our first step was to investigate the effect of query expansion (the process of reformulating a given query to improve retrieval of information) and to observe how users react to conditions where suggestions are actively provided during mind-mapping. For this, we implemented an preliminary interface to record the usage of suggestions retrieved from ConceptNet [42] and conducted a preliminary study using this interface.

### 3.1 Query-Expansion Interface

The idea behind our interface is based on query expansion enabled by ConceptNet. In comparison to content-retrieval analysis (Wiki) or lexical-semantic databases such as WordNet [52], ConceptNet allows for leveraging the vast organization of related concepts based on a diverse set of relations resulting in a broader scope of queries. Using this feature of ConceptNet, we developed a simple web-based tool for query-expansion mind-mapping (QEM, Figure 1, Figure 2) wherein users could add nodes (words/phrases) and link them together to create a map. For every new word or phrase, we used the top 50 query results as suggestions that the users could use as alternatives or additional nodes in the map. Our hypothesis was that ConceptNet suggestions would help users create richer mind-maps in comparison to pen-paper mind-mapping.

### 3.2 Evaluation Tasks

We designed our tasks for (a) comparing pen-paper mind-mapping and QEMs with respect to user performance, preference, and completion time and (b) to explore how the addition of query-based search affects the spanning of ideas in a typical mind-map creation task. Each participant was asked to create two mind-maps, one for each of the following problem statements:

- *Discuss the problem of different forms of pollution, and suggest solutions to minimize them*: This problem statement was kept generic and conclusive, and something that would be typically familiar to the target participants, to compare the creation modalities for simple problem statements.

- *Modes of human transportation in the year 2118*: The intent behind this open-ended problem statement was to encourage users to explore a variety of ideas through both modalities, and observe the utility of query based mind map tools for such problem statements.

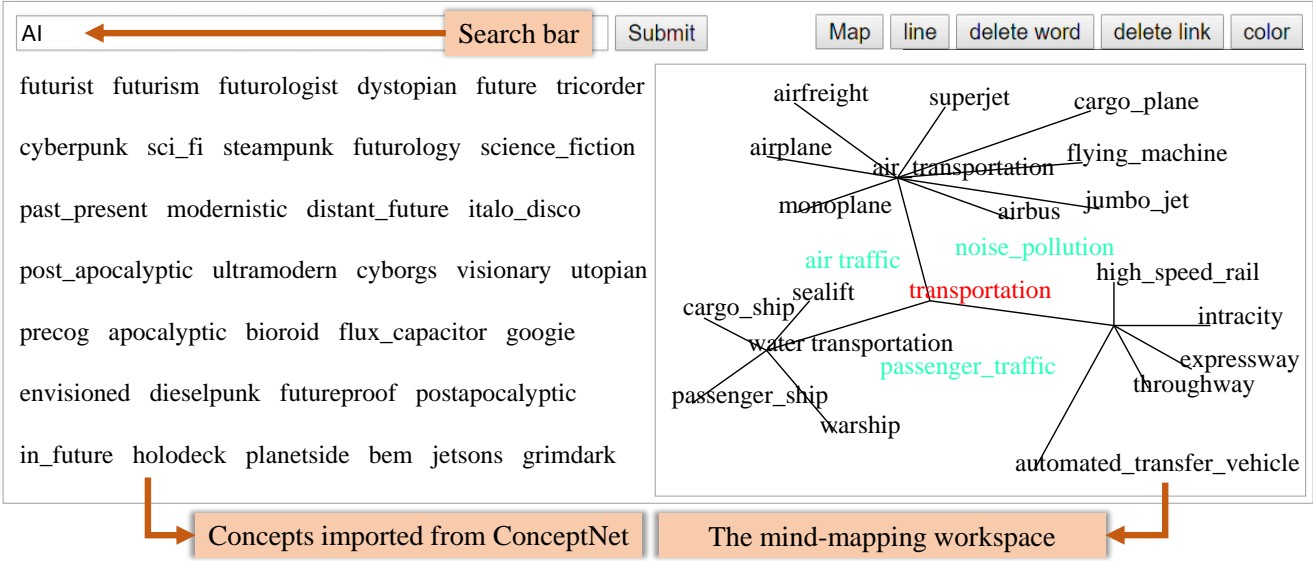

Figure 1: Screenshot of user interface of QEM

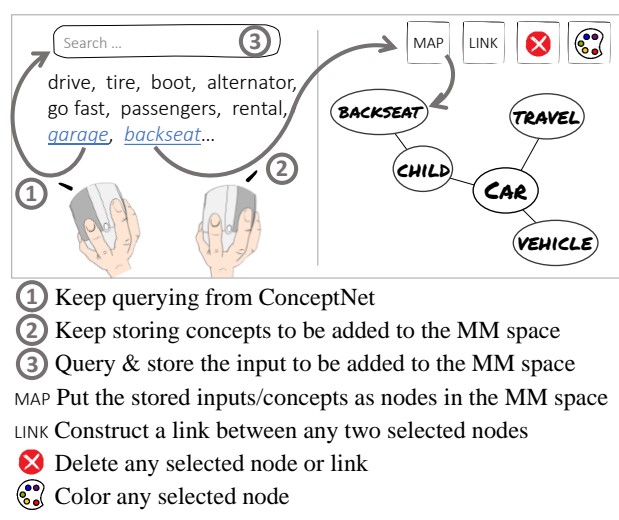

① Keep querying from ConceptNet
② Keep storing concepts to be added to the MM space
③ Query & store the input to be added to the MM space
MAP Put the stored inputs/concepts as nodes in the MM space
LINK Construct a link between any two selected nodes
❌ Delete any selected node or link
🎨 Color any selected node

Figure 2: Illustration of QEM workflow ("MM" stands for "mind-mapping")

The topics for the problem statements were selected to provide users with familiar domains while also leaving scope for encouraging new ideas from the participants.

### 3.3 Participants

We recruited 18 students (10 male, 8 female) from engineering majors between 18 to 32 years of age. Of these, 6 participants were familiar with the concept of mind-maps (with a self-reported score of 4 on a scale of 10). We conducted a between subjects study, where 9 participants created mind-maps for a given problem statement using QEM (Figure 1), and the remaining 9 using provided pen and paper.

### 3.4 Procedure

The total time taken during the experiment varied between 30 to 35 minutes. Participants in the QEM group were first introduced to the interfaces and were encouraged explore the interface. Subsequently, the participants created the mind-map for the assigned problem. They were allowed a maximum of 10 minutes for one problem

statement. Finally, on completion, each participant answered a series of questions in terms of ease of use, intuitiveness, and effectiveness of the assigned mind-map creation modality.

### 3.5 Key Findings

#### 3.5.1 User Feedback

We did not find consensus regarding self-reported satisfaction with the mind-maps created by participants in pen-paper mind-mapping. Moreover, while pen-paper mind-mapping participants agreed that the time for map creation was sufficient, nearly 50% did not agree with being able to span their ideas properly. On the other hand, 90% QEM participants reported that they were satisfied with their resulting mind-maps. Over 80% of the QEM participants agreed to be able to easily search for related words and ideas, and add them to the mind-map. In the post study survey, QEM users suggested adding features such as randomizing the order of words searched for, ability to query multiple phrases at the same time, and ability to search for images. One participant mentioned: *"The interface wasn't able to do the query if put a pair of words together or search for somebody's name viz. Elon Musk"*.

#### 3.5.2 Users' Over-dependency on Query-Expansion

As compared to pen-paper mind-mapping, we observed two main limitations in our query-expansion workflow. First, the addition of a new idea required the query of the word as we did not allow direct addition of nodes in the mind-map (Figure 2). While we had implemented this to simplify the interactions, this resulted in a break in the user's flow of thought further inhibiting diversity (especially when the digital tool is able to search cross-domain and provide a big database for exploring). Second, we observed that users relied heavily on search and query results rather than externalizing their personal views on a subject. Users simply continued searching for the right keyword instead of adding more ideas to the map. This also increased the overall time taken for creating maps using query-expansion. This was also reported by users with statements such as: *"I relied a lot on the search results the interface gave me"* and *"I did not brainstorm a lot while creating the mind map, I spent a lot of time in finding proper terms in the search results to put onto the mind map"*.

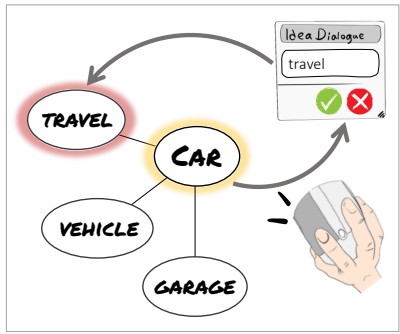 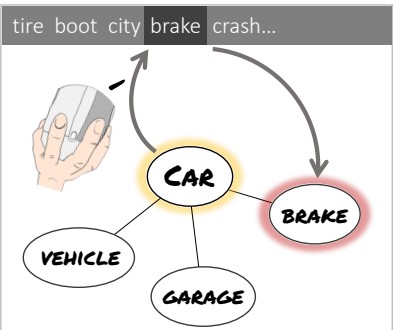 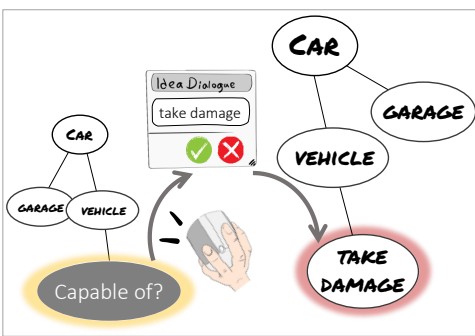

| (a) Direct user input | (b) Input based on suggestions | (c) Input based on response to cue |

Figure 3: Illustration of three mechanisms to create a new node. (a) user double-clicks a node and enters a text entry to create a new child node. (b) single right-clicking an existing node allows the user to use top suggestions from ConceptNet to create a new child node. (c) user double-clicks a cue node to response for a new node addition. Yellow shade denotes selected node; red shade denotes newly created node.

## 4  PHASE II: QCUE

Motivated by our preliminary study, the design goal behind *QCue* is to strike a balance between idea expansion workflow and cognitive support during digital mind-mapping. We aim to provide computer support in a manner that stimulated the user to think in new directions but did not intrude in the user's own line of thinking. The algorithm of generating computer support in *QCue* was developed based on the evolution of the structure of the user-generated map over time to balance the breadth and depth of exploration.

### 4.1  Workflow Design

*QCue* was designed primarily to support divergent idea exploration in ideation processes. This requires an interface that would allow for simple yet fast interactions that are typically natural in a traditional pen-paper setting. We formulate process of mind-mapping as an iterative two-mode sequence: generating as many ideas as possible on a topic (breadth-first exploration), and choosing a smaller subset to refine and detail (depth-first exploration). We further assume our mind-maps to be strictly acyclic graphs (trees). The design of our workflow is based on the following guiding principles:

- In the initial phases of mind-mapping, asking questions to the user can help them externalize their assumptions regarding the topic, stimulate indirect relationships across concepts (latent relations).

- For exploring ideas in depth during later stages, suggesting alternatives to the use helps maintain the rate of idea addition. Here, questions can further help the user look for appropriate suggestions.

### 4.2  Idea Expansion Workflow

We provided the following interactions to users for creating a mind-map using *QCue*:

- *Direct user input:* This is the default mode of adding ideas to the map wherein users simply double-click on an existing node ($n_i$) to add content for its child node ($n_j$) using an input dialog box in the editor workspace. A link is created automatically between $n_i$ and $n_j$ (Figure 3(a)). This offers users minimal manipulation in the construction of a tree type structure.

- *Asking for suggestions:* In situations where a user is unclear about a given direction of exploration from a node in the mind-map, the user can explicitly query ConceptNet with the concerned node (right-click on a node to be queried). Subsequently, we extract top 10 related concepts (words and phrases)

from ConceptNet and allow users to add any related concept they see fit. Users can continuously explore and expand their search (right-clicking on any existing node) and add the result of the query (Figure 3(b)).

- *Responding to cues: QCue* evaluates the nodes in the map and detects nodes that need further exploration. Once identified, *QCue* automatically generates and adds a *question* as cue to user. The user can react to this cue node (double-click) and choose to either answer, ignore, or delete it. Once a valid (non-empty) answer is recorded, the interface replaces the clicked node with the answer (Figure 3(c)).

- *Breadth-vs-depth exploration:* Two sliders are provided on the *QCue* interface to allow adjustment of exploratory directions guided by the cues (Figure 4(a)). Specifically, users can use the sliders to control the position of newly generated cues to be either breadth or depth-first anytime during mind-mapping.

### 4.3  Cue Generation Rationale

There are three aspects that we considered to design our cue-generation mechanism. Given the current state of a mind-map our challenge was to determine (1) *where* to generate a cue (which nodes in the mind-map need exploration), (2) *when* a cue should be generated (so as to provide a meaningful but non-intrusive intervention) and (3) the *what* to ask the user (in terms of the actual content of cue). To find out *where* and *when* to add cues, we draw from the recent work by Chen et al. [13] that explored several algorithms for computer-generated ideas. One of their algorithmic iterations — which is of particular interest to us — involves using the temporal and topological evolution of the mind-map to determine which nodes to target. However, this approach is rendered weak in their work because they modeled the mind-mapping process as a sequential game with each player (human and computer) takes turns. In our case, however, this is a powerful idea since the human and the intelligent agent (AI) are not bound by sequential activity —- both work asynchronously. This also reflects from our core idea of using computer as a facilitator rather than a collaborator. Based on these observations we designed our algorithm to utilize the topological and temporal evolution of a given mind-map in order to determine the potential nodes where we want the user to explore further. For this, we use a strategy similar to the one proposed by Chen et al. [13] that uses two penalty terms based on the time elapsed since a node was added to the mind-map and it's relative topological position (or lineage) with respect to the central problem.

Tesnière [69] note that *continuous thoughts can only be expressed with built connections*. Tesnière was originally describing this idea

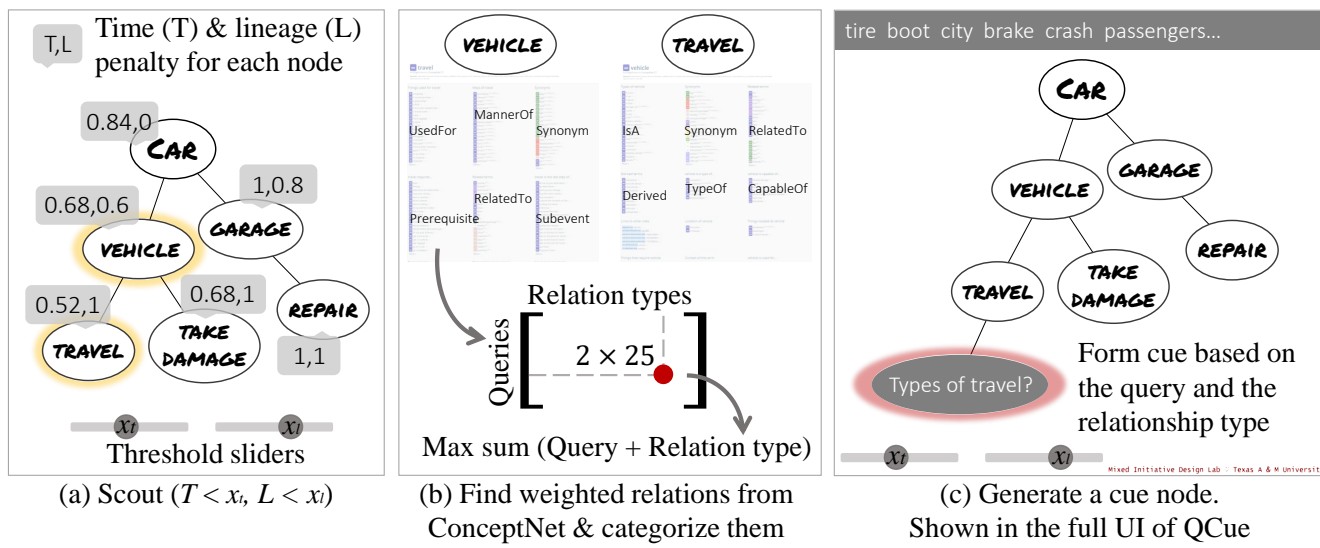

Figure 4: Illustration of the cue generation algorithm using retrieved weighted relations through ConceptNet Open Data API [66]. Yellow shade denotes computer selected potential node; red shade denotes computer generated cue. This algorithm is executed at regular intervals of 2 seconds. The user interface of QCue is illustrated in (c).

in the context of linguistic syntax and how the mind perceives words not in isolation (as they appear in a dictionary) but in the context of other words in a sentence. It is the sentence that provides the connection between its constituent words. This is our driving guideline for composing the content of a cue. Specifically, we observe that the basic issue faced by users is not the inability to create individual concepts but the difficulty in contextualizing broad categories or topics that link specific concepts. Here, we draw from works that identify semantic relations/connections between concepts to build human-like computer systems [54] and perform design synthesis [39]. We further note that the most important characteristic of mind-maps is their linked structure that allows users to associate and understand a group of concepts in a short amount of time. Therefore, our strategy for generating cue content is to simply make use of semantic *relationship types* already provided in ConceptNet. Our rationale is that providing relationship instead of concept-instances will assist the user in two ways: (1) help them think broadly about the problem thereby assisting them in generating much higher number of instances, and (2) keeping a continuous flow of thoughts throughout the creation process. Specifically, we developed our approach by taking the provided 25 relationship categories along with the weighted assertions from ConceptNet into consideration. Note that we did not take all relations from ConceptNet (34 in total) because some may be too ambiguous to users such as *RelatedTo*, *EtymologicallyDerivedFrom*, *ExternalURL*, etc. The algorithm is detailed in the following sections.

### 4.3.1 Time Penalty

Time penalty ($T$) is a measure of the inactivity of a given node in the map. It is defined as the time elapsed since last activity (linked to a parent or added a child). For a newly added node, the time penalty is initialized to 1 and reduced by a constant value ($c$) at regular intervals of 2 seconds. The value of $c$ was determined experimentally (see section 4.4 for details). Once the value reaches 0, it remains constantly at 0 thereafter. Therefore, at any given instance, time penalty ranges from 0 to 1. A default threshold for time penalty was set and adjustable for users by using the provided slider on the *QCue* interface. Users can perform *breadth-first exploration* on ideas that have been recently visited by increasing the threshold value. Given the initial condition $T(n_i) = 1.0$, we compute the time penalty of any

node $n_i \in N_M$ at every interval $\Delta t$ as $T(n_i) \rightarrow \max(T(n_i) - c, 0)$.

### 4.3.2 Lineage Penalty

Lineage penalty ($L$) is a measure of the relative depth of nodes in a given mind-map. It is defined as the normalized total count of children of a given node. Each node has a lineage weight ($x_i$) that equals to 0 upon addition. For the addition of every child node, this weight is increased by 1 ($x_i \leftarrow$ number of children of $n_i$). To compute the lineage penalty for every node, all these weights are normalized (ranges from 0 to 1) and then subtracted by one ($L(n_i) = 1 - x_i/max(x_i)$). Therefore, lineage penalty is 1 for leaf nodes and 0 for the root node, and ranges from 0 to 1 for the others. *QCue*'s support based on this can help *exploration towards leaf nodes*. Note that we give equal importance to all nodes at a given depth of the mind-map. The goal is to determine *where* to generate a cue based on the evolving topology of the maps (acyclic directed graph).

### 4.3.3 Cue Generation using ConceptNet

Given any state of a mind-map, there are three primary algorithm steps that are needed for generating cues in the form of questions using the ConceptNet semantic network. First, *QCue* scouts out a good location (node) to facilitate exploration using the two penalties. Subsequently, the spotted nodes are queried from ConceptNet to retrieve corresponding weighted relations for *content determination*. Finally, based on the determined content, *QCue* generates a cue node to ultimately guide the user and help expand the idea space during mind-map creation.

- *Scouting:* For every node in the current state of a mind-map, we compute its *time penalty* and *lineage penalty*. Then, based on the current adjusted thresholds ($x_t, x_l$) where $x_t$ and $x_l$ denote thresholds for time and lineage penalty respectively, *QCue* spots potential nodes ($N_E$) for exploration. Specifically, if $T(n_i) < x_t$ or $L(n_i) < x_l$ then $N_E \leftarrow N_E \cup \{n_i\}$ (Figure 4(a)). If no node is within the thresholds, all nodes in the current mind-map are considered as potential nodes.

- *Content determination:* In this step, we further query the spotted nodes ($N_E$) from ConceptNet. A list of query results containing weighted relations is retrieved for each potential node

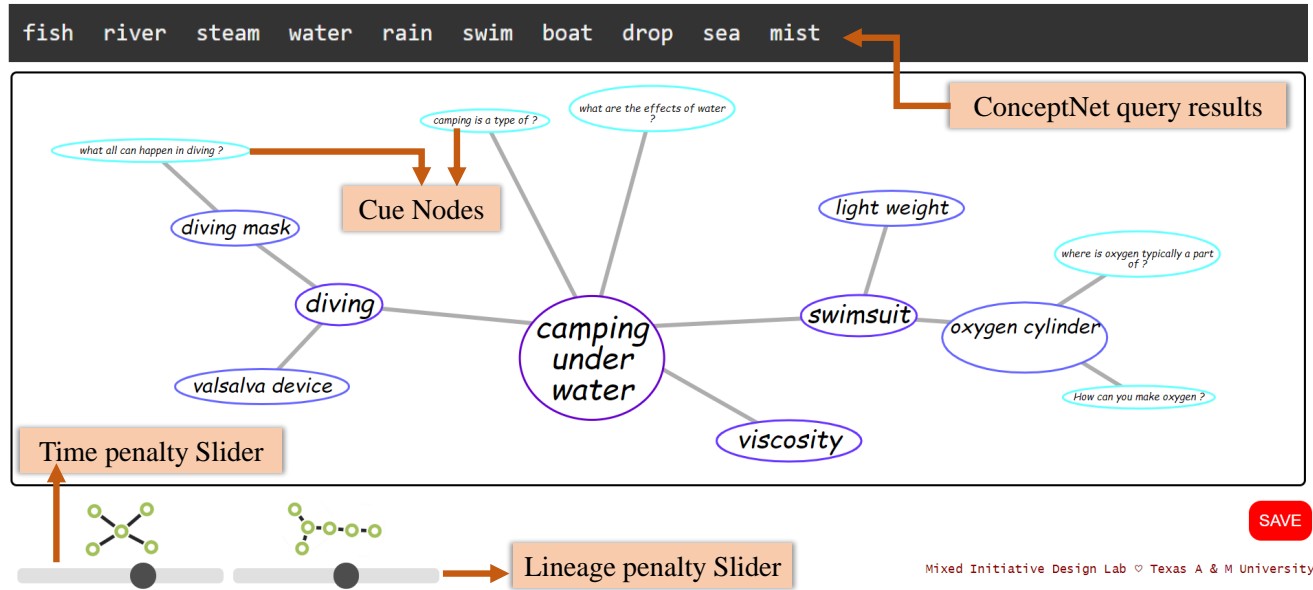

Figure 5: Screenshot of user interface of QCue

(Figure 4(b)). In order to find the node which has the maximum potential of associative capability, we subdivide each list categorically based on the 25 relationship types provided by ConceptNet. Subsequently, we select one subdivision which has the highest sum of relation weights (weights provided by ConceptNet), and use it as basis for a new cue's content (Figure 4(b)). Note that if a subdivision has been used to generate a cue node, it will be removed from future selection pool. For example, *TypeOf* can not be selected again for generating a cue node for *travel* (Figure 4(c)).

- *Cue generation:* Using the selected subdivision from *content determination*, *QCue* formulates a new cue based on fixed templates (Figure 4(c)). To avoid repetition of cues generated during mind-map creation, we specifically construct at least three templates (combinations of query + verb + relationship type) for each relationship category provided by ConceptNet. Example cues based on a query — *knife* — and a relationship type — *CapableOf* — are as follows: *"What can knife do?"*, *"What is knife capable of doing?"* and *"Which task is knife capable of performing?"*.

### 4.4 Implementation

Our *QCue* interface is a Javascript web application that runs entirely on the browser using NodeJS and D3JS (Figure 5). We incorporated JSON-LD API (Linked Data structure) offered by ConceptNet in our interface. The nodes of ConceptNet are words and phrases of natural language. Each node contains an edge list which has all the relations such as *UsedFor* stored in *rel* with its corresponding weight stored in *weight*, and a human-readable *label* stored in *start* and *end*. As the user queries a word or phrase in natural language (as one node), we search for all the relations in this node (filtered in English) and extract the non-repetitive human-readable labels out.

On the *QCue* interface, users can spatially organize ideas in the mind-map by dragging ideas with forced-links around the editor workspace. Such force-directed layout produces an aesthetically pleasing graph while maintaining comprehensibility, even with large dataset. Users are also allowed to adjust sliders to shape their exploration preferences by either wider or deeper. *QCue* employs a listener function to run the cue generation algorithm at fixed intervals

of 2 seconds. We also developed an web-based interface for TMM which is essentially the same as *QCue* but without any computer support (cues and queries).

- *Data format and storage:* Each mind-map is stored in a local folder with a distinct user ID. To store the structure of a mind-map, we defined a JavaScript prototype containing nodes, links, timestamps and other appearance data (e.g. color, size, font etc.). We can regenerate a mind-map by importing the file data into *QCue*. Videos of the mind-maps are also stored within the respective folders to be used in further analysis.

- *Choice of penalty and threshold:* To find an appropriate default value for the constant $c$ in *time penalty* and the thresholds for the two penalties, we conducted several pilot studies (Section 5.1) to observe how people mind-map in a regular setting (TMM) and how people get acquainted with *QCue*. The final assignments are: c = 0.08, $x_t$ & $x_l$ = 0.6 when t = 0.

### 5 EVALUATION METHODOLOGY

### 5.1 Pilot Study

We conducted a pilot study with 12 participants where our intention was to observe (1) how users react to the cue-query workflow, (2) determine ideas and problem statements that could serve as our evaluation tasks, and (3) determine appropriate initial parameters (such as lineage and time thresholds). In order to observe user's thinking process while creating a mind-map, we designed four different problem statements namely, *pollution*, *toys in the future*, *camping underwater*, and *wedding on space station*. We encouraged the users to explore the basic idea, cause, effect and potential solutions of the given problem statement.

Participants were both surprised as well as interested in topics such as *weddings on space station* and *underwater camping*. Specifically, for open-ended topics, they indicated a need for time to prepare themselves before beginning the mind-mapping task. For topics such as *pollution* and *toy*, they showed immediate inclination toward starting the session. Since we wanted to test the robustness of our algorithm with respect to the topic given, we decided to conduct the user study with two topics of opposite extremes. Namely,

| Condition | Structure (1-4) | Exploratory (1-4) | Communication (1-4) | Extent of coverage (1-4) | Quantity (raw) | Variety (0-1) | Novelty (0-1) |
|---|---|---|---|---|---|---|---|
| TMM T1 | 2.29 | 2.42 | 2.38 | 2.25 | 31 | 0.5 | 0.125 |
| TMM T2 | 2.54 | 2.5 | 2.25 | 2.5 | 34 | 0.48 | 0.12 |
| *QCue* T1 | 3.29 | 3.29 | 2.75 | 2.79 | 38 | 0.66 | 0.19 |
| *QCue* T2 | 2.63 | 2.54 | 2.29 | 2.58 | 41 | 0.61 | 0.17 |
| Average TMM | 2.42 | 2.46 | 2.31 | 2.38 | 32.5 | 0.49 | 0.12 |
| Average *QCue* | 2.96 | 2.92 | 2.52 | 2.69 | 39.5 | 0.63 | 0.18 |

Figure 6: Table of average ratings for each metric by four user conditions: TMM, QCue with T1 and T2. On a scale of 1 to 4: 1 – Poor, 2 – Average, 3 – Good, 4 – Excellent.

*pollution* (**T1**) - a seemingly familiar topic and *underwater camping* (**T2**) - a more open-ended topic that is uncommon to think about.

## 5.2 Participants

In the user study, we recruited 24 undergraduate and graduate students from all across a university campus. Our participants came from engineering, architecture, and science backgrounds and were within the age range of 19-30 years. Six (6) participants had prior experience with creating mind-maps. For those who had no experience with mind-mapping, we prepared a short presentation about the general spirit and principles of the technique, and provided them an additional 5 to 10 minutes to practice. We conducted a between-subjects study to minimize learning effects across conditions, where 12 participants created mind-maps for a given topic using TMM, and the remaining 12 using *QCue*.

## 5.3 Tasks

In total, across the two experimental conditions, 24 participants created 48 mind-maps — one for each central topic. The total time taken during the experiment varied between 30 and 40 minutes and the order of the two central topics were randomized across the participants. After describing the setup and the purpose of the study, we described the features of the assigned interface and practically demonstrated its usage. For each participant and the mind-mapping task, we recorded a video of the task, the completion time, and the time-stamped ideas generated by the users for each mind-map. Each participant performed the following tasks:

- *Practice*: To familiarize themselves with the interaction of the assigned interface, the participants were given a brief demonstration of the software and its function. They are allowed to practice the interface for 5 to 10 minutes, with guidance when required.

- *Mind-mapping with **T1** & **T2***: Participants were asked to create mind-map using the assigned interface. The duration of mind-mapping session was 10 minutes for each central topic. Participants were encouraged to explore the central topic as fulfill as they could. The workspace was cleared after completion of each mind-map.

- *Questionnaire*: Finally, each participant answered a series of questions regarding their exploration of central topic before and after the creation of each mind-map, perception of each of the interfaces in terms of ease of use, intuitiveness, and assistance. We also conducted post-study interviews to collect open-ended feedback regarding the experience.

## 5.4 Metrics

Mind-maps recorded during the study were de-identified. The designed metrics assessed all ideas generated in each mind-map based on four primary aspects: **quantity, quality, novelty and variety** [45, 63]. The **quantity** metric is directly measured as the total number of nodes is a given mind-map. The **variety** of each mind-map is given by the number of idea categories that raters find in the mind-map, and the **novelty** score is a measure of how unique are the ideas represented in a given mind-map [12, 45]. For a fair assessment of the **quality** of mind-maps for both central topics, we adapted the mind-map assessment rubric [1, 12] and the raters evaluated the mind-maps based on the four major criteria: **structure, exploratory, communication and extent of coverage**[1]. These metrics are commonly used to evaluate ideation success in open-ended design tasks [10].

Here, we would like to point out to similar metrics that have been used in HCI literature on creativity support. For instance, Kerne's elemental metrics [37] for information-based ideation (IBI) are adapted from Shah's metrics [63]. While the metrics we chose have been used in previous mind-mapping studies, they also have some connection with creativity-support index (CSI) [9, 14] and ideational fluency [37] (for example, holistic IBI metrics are similar to the "structure" metric and our post study questions are functionally similar to CSI tailored for mind-mapping).

## 5.5 Raters

Two raters were recruited for assessing the mind-maps created by the users. These raters were senior designers in the mechanical engineering design domain, having had multiple design experiences during their coursework and research life. The raters selected were unaware of the study design and tasks, and were not furnished with information related to the general study hypotheses. The 48 mind-maps created across both interfaces were presented to each rater in a randomized order. For every mind-map assessed, the raters evaluate them on a scale of 1-4 based for each criteria discussed above. Every created mind-map is then assigned a score from a total of 16 points, which is used further for comparing the quality with respect to other mind-maps.

For a given central topic, the evaluation depends on knowledge of the raters and their interpretation of what the metrics mean. In our study, two inter-raters independently perform subjective ratings of every idea/concept in a mind-map. This evaluation technique has the advantage of capturing aspects of creative work that are subjectively recognized by raters, but are difficult to define objectively. After the independent evaluation by the two raters, the ratings from the two raters were checked for consensus.

---

[1]Please refer to the literature for detailed explanation of these metrics

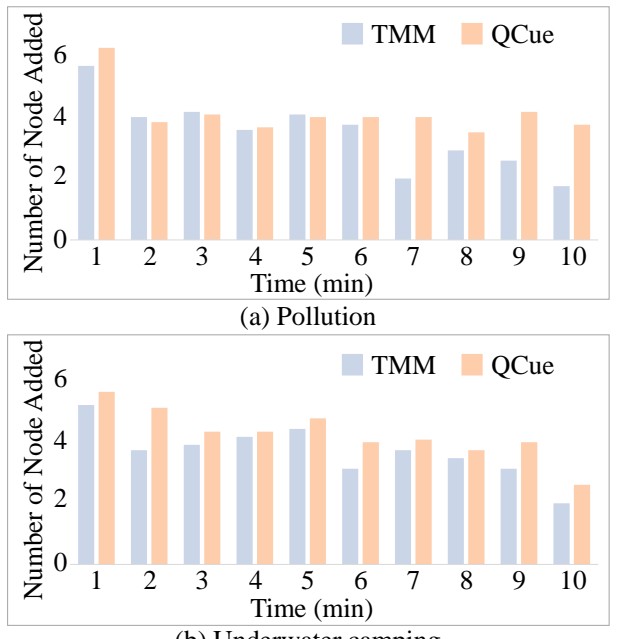

Figure 7: General trends on how users generating ideas towards different topics (T1 and T2) during TMM and QCue. Each bar represents an average count of the total nodes in the given time frame (per 1 minute).

## 6 RESULTS

### 6.1 Ratings for User-Generated Mind-Maps

For metrics admitting integer values (structure, exploratory, communication and extent of coverage), we calculated the Cohen's kappa for ensuring inter-rater agreement. The Cohen's kappa value was found to be between the range of $0.4 - 0.6$ showing a moderate inter-rater agreement level [15]. For metrics admitting real/scalar values (variety and novelty), we calculated the Pearson's correlation coefficient to find the degree of correlation between the raters' ratings. This correlation coefficient was found to be close to 0.8 which indicates an acceptable range of agreement [15].

Overall, the ratings for *QCue* were relatively higher than TMM across all metrics (Figure 6). Two-way ANOVA was conducted with two factors of comparison: (1) the choice of topic (*pollution* or *underwater camping*) and (2) the choice of interface (*QCue* or TMM). Although the data for certain metrics were non-normal, we proceeded with ANOVA since it is resistant to moderate deviation from normality. The mean ratings for structure were higher for QCue (2.96) in comparison to TMM (2.42, p-value 0.007). Similarly the mean scores for the exploratory metric is also higher for *QCue* (2.92) with respect to TMM (2.46, p-value 0.008). This suggests that the mind-maps created using *QCue* were relatively more balanced (in depth and breadth) and more comprehensively explored. Further, we recorded a better variety score in *QCue* (0.49) relative to TMM (0.63, p-value 0.009). Finally, we also recorded a larger number of nodes added in *QCue* (39.5) relative to TMM (32.5, p-value 0.048). These observations indicate that the cue-query mechanism assisted the users in (1) exploring diverse aspects of the given topic and (2) making non-obvious relationships across ideas.

We also carried out a main effect analysis (one-way ANOVA) between *pollution* and *underwater camping* independently for TMM and *QCue*. While the difference in the outcome was not pronounced in TMM, a significant difference was found across topics in the structure ($p = 0.01$) and exploratory ($p = 0.002$) metrics for *QCue*. This suggests that the *QCue* workflow is dependent on the type of

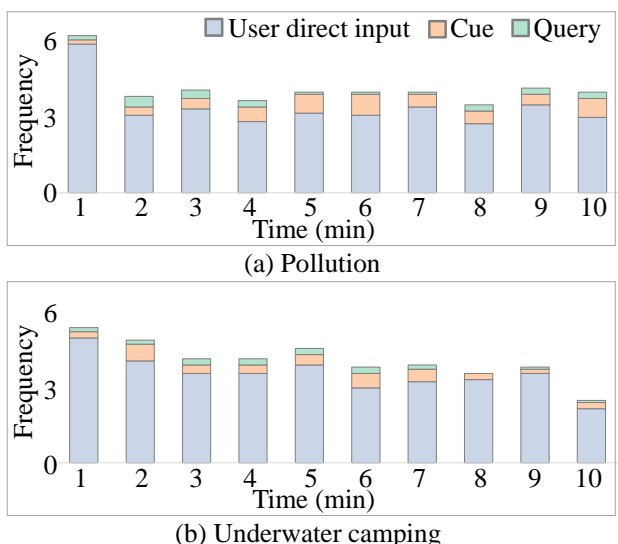

(a) Pollution

(b) Underwater camping

Figure 8: Comparison of trends on how QCue users generating ideas towards T1 and T2 using the three modes (user direct input, cue node response and query) stacked one above the other. The frequencies are averaged across the 12 users.

the central topic explored. The overall ratings are higher in *QCue* for *pollution* (for example in Figure 6, mean structure value increases to 3.29 from 2.29 in *pollution*).

### 6.2 Temporal Trend for Node Addition

In general, the rate of node addition decreased over time in the TMM workflow regardless of the topics. For *QCue*, the node addition rate was comparatively steady indicating that cues and queries helped sustain user engagement for exploration during even later stages of the tasks (Figure 7).

While there are three modes for node addition using *QCue*, as expected, the number of cues and queries used depended on users' familiarity with the central topic in the tasks. Overall, we observed that the users tended to ask for queries in the first few minutes of mind-mapping, and proceed with the usage of cue nodes in the middle stages of the given time (Figure 8). For *pollution*, the number of answered cue nodes increases with time. Specifically, users appreciated cues between the 5 and 6 minutes mark for *pollution*. For *underwater camping*, we noticed an increasing amount of the cue nodes answered specifically in the 2 and 6 to 7 minutes mark. This indicates two primary usage of cues. First, when the users have explored their prior knowledge of the topic and reach an impasse during the middle stages of mind-mapping (5 to 7 minutes mark in our case), cues help them reflect on the existing concepts and discover new relationships to generate ideas further. Second, for open-ended problems such as *underwater camping*, cues helped users in exploring different directions of exploration around the central idea in the beginning. This impacted the exploration of ideas in the later stages of the task. On the other hand, surprisingly, we found that the percentage of the number of nodes added from query mode is lower than the cue mode. This suggests that users were generally more engaged when they were actively involved in the cycle of exploration and reflection based on cues in comparison to receiving direct answers provided by query.

### 6.3 User Feedback: Cue vs Query

To help us evaluate the effectiveness of our algorithm, the participants filled out a questionnaire after creation of each mind-map.

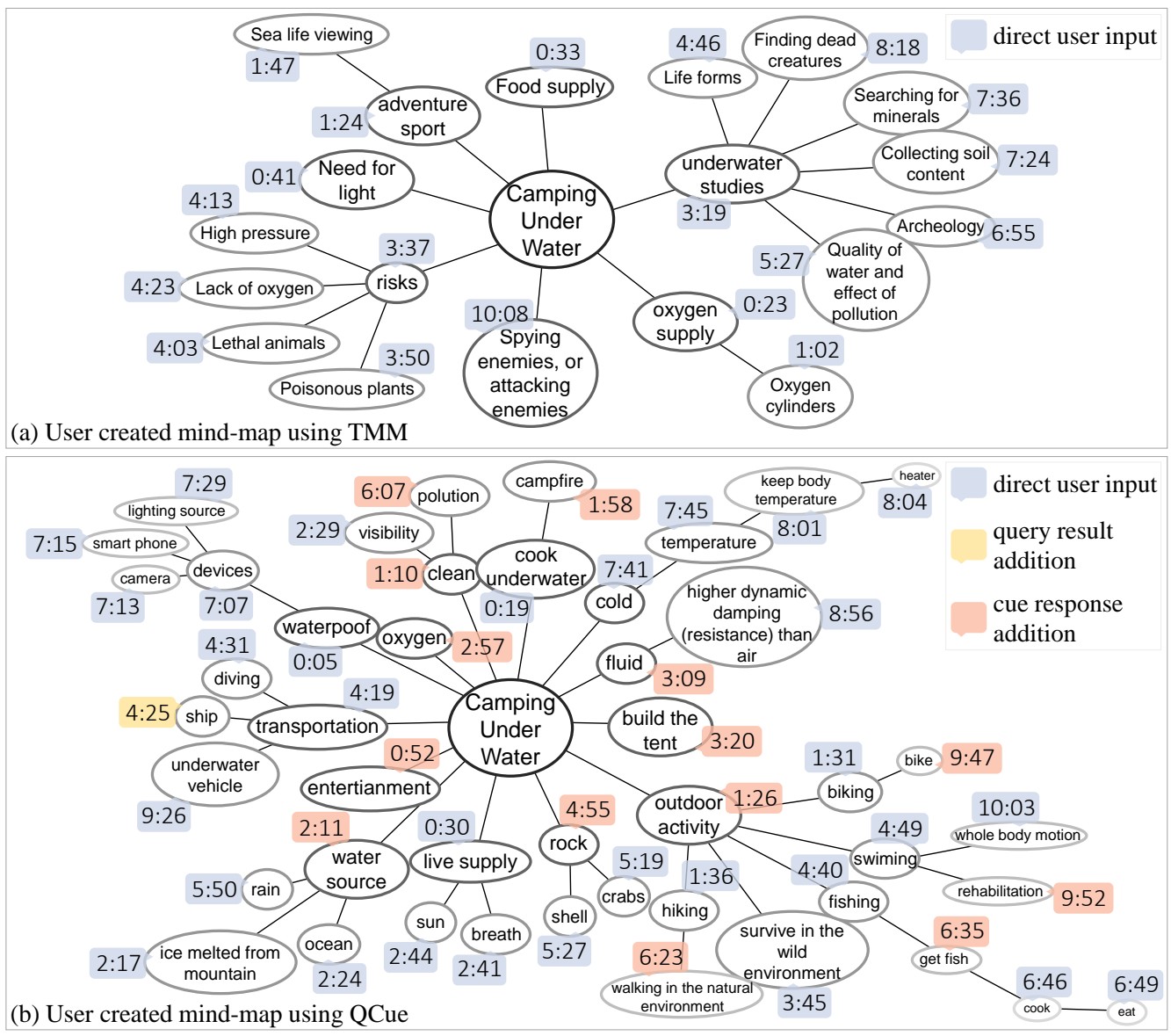

Figure 9: Two user created mind-maps with underwater camping (T2) as central topic using (a) TMM and (b) QCue. The label represents timestamp upon node addition and type of addition.

We also encouraged the participants to give open-ended feedback to support their rating.

There was a mixed response from the users for asking *whether the cues were useful in the process of mind-mapping*. Around 60% of the users agreed that the cues helped them to develop new lines of thoughts at the right time. One user stated, *"Questions (or cues) were helpful at the point when you get fixated. They give you other dimensions/ideas to expand your thought"*. The remaining stated that they do not find the cues helpful because they already had ideas on how to develop the mind-map. *"I felt like the questions (or cues) would make me lose my train of thought"*. Users who found it difficult to add to existing ideas in the mind-map, used the cues and queries extensively to build and visualize new dimensions to the central idea. These users felt that the cues helped them to reach unexplored avenues: *"I started with a particular topic, and ended at a completely unrelated topic. It enabled me to push my creativity limits further"* .

For the usage of queries, above 80% of users agreed that queries were useful regardless of the topics. For *underwater camping*, 20% of the users who disagreed, suggested that the system should include queries that were more closely linked to the context of the central idea. Specifically, a user stated: *"Some suggestions (or queries) under certain context might not be straight forward"*.

What is interesting to note here is that while we received mixed responses in the cues and overly positive responses on queries, we also recorded higher number of user interactions with cues than queries. The likely explanation for this seeming contradiction is that it is easy to answer a cue than looking for a suggestion that fits the user's need at a given instance. Second, querying a suggestion also would mean that the user was clear in what they wanted to add. However, this clarity ultimately resulted in users directly adding the node manually. Therefore, we believe that the users tacitly inclined toward answering to the cues generated by our system.

### 6.4 User Feedback: QCue as a Workflow

In comparison to TMM, users who used *QCue* performed more consistently during creation of mind-maps — the frequency of generating new nodes was comparatively steady throughout the process. As one user stated: *"the questions helped me to create new chain of thoughts. I might not have the answer for the question (or cues) directly, but it provided new aspects to the given idea. Especially for underwater camping"*. One user with negligible experience in brainstorming, shared her excitement: *"I was fully engaged in the creation process. I was expecting questions from all different angles"*. On the other hand, we also found that *QCue* users kept generating new directions of ideas with respect to the central topic even after the initial creation phase, where TMM users tended to focus on fixed number of directions (Figure 9). This indicates the capability of *QCue* — problems co-evolved with the development of the idea space during the mind-mapping process.

## 7 DISCUSSIONS

### 7.1 Limitations

There are two main limitations in this work. First, a majority of the recruited users had little to no experience in mind-mapping. While this allowed us to demonstrate the capability of *QCue* in guiding novices to explore problem spaces, we believe that including expert users in our future studies can help us (1) understand how differently they perform using this workflow and (2) lead to a richer discussion on how expertise can be transferred to our system toward better facilitation. Second, one of the key challenges we faced was the lack of robust methodology for determining the effect of cue-based stimulus during mind-mapping (how users may have used cues and queries without explicitly using them to add nodes). While we characterize it on the basis of the number of cues answered and the number of suggestions assimilated directly in the mind-map, we believe that a deeper qualitative study on the mind-mapping process can reveal valuable insights. We plan to conduct such an analysis as our immediate next step.

### 7.2 Cue & Query Formulation

One of the challenges we faced in our implementation of cue generation was grammatically and semantically effective formulation of the questions themselves. Recently, Gilon et al. [28] demonstrated a design-by-analogy workflow using ConcpetNet noting the lack of domain specificity to be an issue. In this regard, there is scope for further investigation of natural language processing methods as well as new databases for construction of cues in specific domains such as engineering design. More importantly, users frequently suggested for context-dependent queries. For problems such as *underwater camping*, this is a challenging task that may need technological advancements in artificial intelligence approaches for generating suggestions and cues based on real-time synthesis of ideas from the information retrieved from a knowledge database. We did preliminary exploration in this direction using a markov chain based question generation method [16]. However, the cues generated were not well-phrased indicating further studies into other generative language models [23].

### 7.3 Cue Representation

The rationale behind providing cues comes from being able to stimulate the user to generate and add ideas. We believe there is a richer space of representations, both textual and graphical, that can potentially enhance cognitive stimulation particularly for open-ended problems. For instance, textual stimuli can be produced through simple *unsolicited* suggestions from ConceptNet (example: *"concept?"*) or advanced mechanisms based on higher level contextual interpretation (e.g. questioning based on second-order neighbors in the ConceptNet graph). From a graphical perspective, the use of visual content databases such as ShapeNet [11] and ImageNet [21]

may lead to novel ways for providing stimuli to users. There are several avenues that need to be investigated in terms of colors, images, arrows, and dimension to reflect personal interest and individuality [8].

## 8 CONCLUSION

Our intention in this research was to augment users' capability to discover more about a given problem during mind-mapping. For this, we introduced and investigated a new digital workflow (*QCue*) that provides cues to users based on the current state of the mind-map and also allows them to query suggestions. While our experiments demonstrated the potential of such mechanisms in stimulating idea exploration, the fundamental take-away is that such stimulation requires a balancing act between intervening the user's own line of thought with computer-generated cues and providing suggestions to the user's queries. Furthermore, our work shows the impact of computer-facilitated textual stimuli particularly for those with little practice in brainstorming-type tasks. We believe that *QCue* is only a step toward a much richer set of research directions in the domain of intelligent cognitive assistants.

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
