# OpenReview forum: "QCue: Queries and Cues for Computer-Facilitated Mind-Mapping"
_graphicsinterface.org/Graphics_Interface/2020/Conference — GI 2020_

### Official Review · AnonReviewer1 · 2019-12-29
**Interesting stuff!**

**Confidence:** 3
**Rating:** 7

**Review:**


This manuscript considers the problem of how to enhance the practice of mindmapping, where people construct a visual representation to support a brainstorming process. The authors' approach is to design a new workflow/tool (called QCue) that can support different aspects of mindmapping through the generation of suggestions (where the suggestions are driven by an online ontology called ConceptNet). The authors evaluate this tool through two phases, the first which considers suggestions for terms to put into the mindmap; the second of which considers the full-blown tool (suggestions for terms in the mindmap; suggestions for new nodes in the mindmap based on relationships). The evaluation suggests that the mindmaps generated with QCue are better than those generated without this tool support. The contribution of this work is in illustrating how to enhance a cognitive brainstorming task with computational support.

The manuscript is tidily written, and provides a nicely motivated contribution to the community. Although I am not an expert in the particular domain of work that the research is focused on, there are some nice ideas in the design of the tool. The introduction is excellent -- it introduces the problem well, describes the approach, the evaluation and provides a clear roadmap for the manuscript. I think the related work section is also pretty good. It does an excellent job of laying out the field, and setting the scene. References seem to be great starting points for others. The only thing that I wondered about was why the digital mind mapping stuff was last, since this was the focus of the manuscript. The linkage between this and the problem exploration and computer-based cognitive support could be made even more explicit (i.e. that digital mind-mapping is an instance of these). The only bummer is that the review of digital mind-mapping tools is a bit on the thin side -- for instance, the two references that describe the problems with digital mind mapping (65,5) are written well before [25], and so it's not clear if these problems have since been resolved.

For the remainder of the manuscript, I provide below some smaller points for the authors to clarify. While I think these are executed reasonably well, there are perhaps some opportunities for the authors to consider in future work, or at least to reflect on in the current manuscript (say, in the discussion). Overall, I think the manuscript should be accepted.

Some larger issues that the authors should address are as follows:

* In phase 2, the authors describe their model of a mindmapping process to include "choosing a smaller subset to refine and detail", yet my interpretation of the algorithm is that it treats all nodes of equivalent depth equally. Thus, the tool support does not account for this in a semantic way; rather, it simply says (through the cues), "Hey, you haven't looked at this for a while," if it is a subtree the user has chosen to ignore. I think this shortcoming of the algorithm should be noted clearly, since the algorithm seems mainly to push for areas of the mindmap has not been worked on for a while. Similarly, when constructing mindmaps, we know that some nodes are not worthwhile progressing further; how can we inform the tool of this? Or, how could the tool account for this in the future?
* The UI for the tool is not illustrated well in the manuscript. While the video does a lot to alleviate this, my belief is that the manuscript should still stand on its own in this regard. I would recommend taking space from Figures 2, 6 as space for illustration or description of the UI.

## Phase 1:

Points of clarification:
* Perhaps providing a clearer picture of the UI for the tool participants used in the query-expansion version would be useful. E.g. Is this what is illustrated in the video? As-is, readers need to guess what the experience was like.
* For the "pen and paper" version, were participants really using a pen and paper?
* 6/18 participants were familiar with mind-mapping; does this mean the other 12 had no experience with mind-mapping? Was there effort to familiarize these participants to the technique? [This is somewhat addressed in the discussion; it may be useful to speculate on how the tool would perform with different populations -- those familiar with mindmapping vs those not familiar with it]
* Was this a within-subjects design? Is it counterbalanced?
* Do the authors consider both prompts to be essentially equivalent, or not?
* This sentence is awkward: "Moreover, while pen-paper mind-mapping participants agreed that the time for map creation was suffi- cient, nearly 50% did not agree with being able to span their ideas properly." -- how do participants know whether they are able to span well or not?
* Awkward (partly because we don't know what the UI looks like): "... we observed two main limitations in our query-expansion workflow. First, the addition of a new idea required the query of the word."

## Phase 2:

Perhaps it would be useful to clarify this point:
* "we subdivide each list categorically based on the 25 relationship types provided by ConceptNet. Subsequently, we select one subdivision which has the highest sum of relation weights and use it as basis for a new cue’s content (Figure 2(b)). " -- what are the relation-weights? does this vary over time? does it vary somehow? or, does "Car" always result in the same subdivision? (where do the relation weights come from??)

## Main Study

* The main effect for the topic type is not clear (as a reader, we don't know why this is important or relevant). It might be useful to provide an explanation or interpretation.
* Figures 4 and 5 are a bit hard to read. For Figure 4, perhaps the error bars (or variance lines) aren't necessary (they obscure the message). For Figure 5, I just am not sure what this means. Are these for both groups of participants? Are they stacked charts? My guess is that they are only for QCue participants (this should be made explicit), and that they are stacked, I think these are reporting on averages across participants. If so, then the variation from minute to minute seems pretty low. I'm not sure how meaningful this chart is.
	- It might instead be useful or interesting to consider what suggestions/queries were made, and whether they were used (or not). Also what about entries that were developed based on those the results of those queries?
* For me, I would have preferred to see a bit more qualitative description of what participants did/how they used the QCue interface. The current minute-by-minute breakdown does not seem to illustrate this well for me.
* The main result comes from expert evaluation. I buy this on face value, but it might be useful to understand the rubric the authors (or the evaluators) used for themselves. For instance, what does a 2, 3 or 4 rating on these metrics mean for them? Is the difference between a 2.1 score and a 2.9 score meaningful?

## Discussion
* It may be worthwhile for the authors to consider whether the end result (of using QCue or TMM) to be different for the participants themselves. While I am not an expert in this space, I seem to recall hearing from some professional development workshop that mindmapping was a good tool to help brainstorm, but the mindmap itself was not a useful artefact in the end -- rather it was the *act of construction* that help to create these "mental structures" that would provide utility later on. If we buy this interpretation, then how can we evaluate how rich the participants' own mental structures of the problem spaces are? (Since, this is the main goal, I would guess -- i.e. it's not really about enhancing the creation of a mindmap artefact; it's really about enhancing one's cognitive understanding of the problem domain).
* Perhaps for future work, might the authors consider an altogether different type of UI presentation? My current read of the cues is that they appear over time (kind of like an alert prompt). Personally, I think I would find this stressful (in the same way that an unread email count in my mailbox bothers me). What if we considered an on-demand workflow -- e.g. when I am stuck, I get to ask the tool, "Hey, where should I work now?"
* Something the manuscript made me think about was how teachers support mindmapping activities among students. It might be interesting (perhaps not for these authors, but in the future) to explore how it is that teachers decide when and how to prompt students that are stuck in their mindmapping activities, and then to understand how the algorithm could be modified to model this type of suggestion-making approach that teachers use.

Overall, I like this manuscript. But, it is still possible to push it further. For instance, the tool is cool, but I am left without a clear direction for future work. Do we think the design was perfect? Is there something we could improve on? Either in the algorithm, or the UI? Or, should the study be improved somehow? Are we convinced of the result? Are there other places we could apply this idea? These things might be interesting points for the authors to consider.

---

### Official Review · AnonReviewer2 · 2020-01-02
**Well-conducted research and easy to read paper**

**Confidence:** 4
**Rating:** 8

**Review:**

This paper presents an approach for assisting people with mindmapping tasks by prompting them to expand particular nodes (to add additional depth), or to consider different aspects of the problem (to add additional breadth). These suggestions are powered by ConceptNet -- an online semantic network of concepts. In addition to the design of an interface to support mindmapping, and the algorithmic approaches to power these features, the paper contributes a evaluative study demonstrating that the QCue approach enabled users to explore diverse aspects of a given topic, and make non-obvious relationships across ideas. The paper also provides insights into future directions for developing this approach.

Overall, I enjoyed reading this paper and I believe it should be accepted. Though the idea of providing support for ideation and brainstorming is not new, I am unaware of other work that has looked at mind-mapping specifically, or the application of semantic networks such as ConceptNet to this purpose. I appreciated the detail with which the paper described the process through which QCue was developed, including a preliminary study, the design of the interface and underlying algorithms, the rationale for how cues are generated, and a detailed evaluation that includes quantitative measures, ratings by experts, and subjective feedback. Moreover, I found the writing and presentation throughout the paper to be generally clear and easy to follow.

Though I'm generally quite positive on this work, I do have a few small criticisms:

First, I felt that the paper could do more to acknowledge and summarize some of the other approaches that have been used to support brainstorming activities, outside of the area of mindmapping. Notable work that comes to mind is Siangliulue et al.'s IdeaHound paper from UIST 2016, which includes a solid review of creativity enhancing interventions, which may contain other relevant work to cite.

Second, I found some of the reporting of questionnaire results to be confusing. In particular, it was not clear the exact question that was asked of participants and what the responses were. For example, on page 3 the paper reads "We did not find consensus regarding self-reported satisfaction with the mind-maps created by participants in pen-paper mind-mapping.", and does not provide further detail than that. It would be good to specify clearly the questions asked (including the wording), and the counts of participants that provided different ratings or responses. This applies to both the preliminary study and to the larger evaluation study.

Finally, there are some minor grammatical and wording issues that could be corrected:

- pg. 1 - "Asking of one question…" - awkward
- pg. 1 - The sentence that begins "We apply this tenet…" is a run-on sentence.
- pg. 2 - "and limit the" should be "and limits the"
- pg. 2 - "during idea generation process" should be "during the idea generation process"
- pg. 2 - "…from note-taking to information integration in areas" - "in areas" can be deleted
- pg. 2 - The sentence beginning "Few works that have considered this idea [7, 25] have…" is awkward and should be rephrased.
- pg. 3 - "18 students (8 females)" - you should report # of males, # of females, and any participants who declined to answer or specified something else, rather than assuming total = males + females
- pg. 4 - "Once a valid answer recorded, the…" - missing word
- pg. 5 - "Tesniere et al [66] note that continuous thoughts can only be expressed with built connections." - I think it's worth unpacking the meaning of this a bit more, for people who aren't familiar with this work.
- pg. 5 - "…at regular intervals of the computational cycle." - I'm not sure what is trying to be expressed by this -- that CPU ticks were used?
- pg. 5 - "node that added to a given node" - awkward
- pg. 5 - "using ConceptNet semantic network" - missing "the"
- pg. 6 - "As user queries a word or phrase in natural language" - "As *the* user queries…"
- pg. 6 - "Section 5.1" - There are no section numbers in this format
- pg. 6 - "On the other hand" - this is informal language, and also you didn't set it up with "On one hand" earlier, so it shouldn't be used.
- pg. 6 - "rather a more abstract topic." - I don't think that it's more abstract, it's just that it isn't something people usually think about.
- pg. 6 - "…and the two central topics were randomized across the participants." - This suggests that each participant was only assigned one topic. Do you mean that the order of the two topics was randomized for each participant?
- pg. 7 - "P M Q" are used for each phase, but then are never referred to again. Unless you have a good reason to create new terminology or assign symbols, it's better to not.
- pg. 8 - "after creation of one mind map" - Why only one and not both? Do you mean "each"?
- pg. 10 - "generating new directions of ideas respected to the central topics" - awkward
- pg. 10 - "The rationale behind cue comes from…" - The rationale behind "providing cues", or "cuing users"

---

### Official Review · AnonReviewer3 · 2020-01-08
**Well-written paper and interesting work**

**Confidence:** 3
**Rating:** 8

**Review:**

This paper presents QCue, a tool to assist mind-mapping through suggested context related to existing nodes and through question that expand on less developed branches, including two studies, a detailed description of the algorithm design, and rater evaluation of their results. The first study explores how users respond to new node ideas suggested by the tool and whether that creates more detailed maps. The second study expands on those findings to balance the depth and breadth of mind maps creation. Both studies compare the new mind mapping tool to digital options without computer assistance. They find that QCue produces more balanced and detailed mind maps and that some mind mapping tasks may be better suited to this type of computer intervention than others.

Overall, this paper is an interesting exploration of a novel area of computer supported brainstorming. The two studies are well-described and designed studies. The level of detail in the algorithm description is a particular strength, giving a clear picture of how it works and why those choices were made. One small point that could be clarified is why a between subjects design was chosen over a counterbalanced within subjects.

Finally, the discussion would benefit from some more general discussion, before the limitations, on the overall findings and what they mean for mind mapping and similar applications moving forward. The results are individually compelling, but what does it mean all together? This research is well-written and a good contribution to the area of brainstorming, and it would be interesting to get more of a complete sense of the results.

---

### Meta-Review · Area_Chair1 · 2020-01-10

**Recommendation:** Accept
**Confidence:** 4

**Metareview:**

This paper received three positive reviews, all of which recommend that the paper be accepted. Based on this, I am recommending 'Accept' as well. Though all of the reviews were positive, they also provided suggestions on how to improve the paper, and strengthen its presentation and contribution. I encourage the authors to take these suggestions to heart, and to try and integrate them into the final version of the paper.

---

### Decision · Program_Chairs · 2020-01-11

Accept